# Current Trends in Molecular Imprinting: Strategies, Applications and Determination of Target Molecules in Spain

**DOI:** 10.3390/ijms24031915

**Published:** 2023-01-18

**Authors:** Idoia Urriza-Arsuaga, Miriam Guadaño-Sánchez, Javier Lucas Urraca

**Affiliations:** 1Independent Researcher, 28007 Madrid, Spain; 2Chemical Optosensors and Applied Photochemistry Group (GSOLFA), Department of Analytical Chemistry, Facultad de Química, Universidad Complutense de Madrid, 28040 Madrid, Spain

**Keywords:** molecularly imprinted polymers, strategies of polymerization, molecular imprinting

## Abstract

Over the last decades, an increasing demand for new specific molecular recognition elements has emerged in order to improve analytical methods that have already been developed in order to reach the detection/quantification limits of target molecules. Molecularly imprinted polymers (MIPs) have molecular recognition abilities provided by the presence of a template molecule during their synthesis, and they are excellent materials with high selectivity for sample preparation. These synthetic polymers are relatively easy to prepare, and they can also be an excellent choice in the substitution of antibodies or enzymes in different kinds of assays. They have been properly applied to the development of chromatographic or solid-phase extraction methods and have also been successfully applied as electrochemical, piezoelectrical, and optical sensors, as well as in the catalysis process. Nevertheless, new formats of polymerization can also provide new applications for these materials. This paper provides a comprehensive comparison of the new challenges in molecular imprinting as materials of the future in Spain.

## 1. Introduction

Molecular imprinting is a process in which selected functional monomers are allowed to self-assemble around a template molecule and are subsequently polymerized in the presence of a crosslinker. After the template extraction, the resulting polymer will have a “memory” of the shape, size, and functional group orientation of the template molecule, allowing for its selective rebinding even in the presence of compounds with a structure and functionality similar to those of the template in complex matrices.

Therefore, molecularly imprinted polymers (MIPs) act essentially as artificial antibodies and are known and applied as antibody mimics for different applications, as excellent molecular recognition properties can be achieved with MIPs for a variety of molecules [1].

MIPs are typically synthesized in a three-step process. In the first step, the functional monomers interact, by covalent or noncovalent interactions, with the template molecule. After the addition of the cross-linker and usually through a radical polymerization (then, the addition of an initiator is also required), the mixture is polymerized in an appropriate solvent. The final step involves the removal of the template molecules resulting in a cross-linked material with selective binding cavities for the target compound, as shown in Figure 1 [2].

### 1.1. Polymerization Models

There are three strategies or models for molecular imprinting depending on the type of interaction between the template molecule and the functional monomers in the prepolymerization mixture [3].

#### 1.1.1. Covalent Bond

This model is based on the covalent interaction between the template molecule and the functional monomer(s). The application of this approach requires an additional step to transform the template molecule into a polymerizable derivative. After the polymerization process, the template molecule is extracted; subsequently, during the recognition process, the covalent bond is formed again (Figure 2) [4].

#### 1.1.2. Semicovalent Model

This model is based on a covalent interaction between the functional monomer and the template in the prepolymerization. However in this approach, the recognition process between the template molecule and the polymer takes place by noncovalent interactions [5].

#### 1.1.3. Noncovalent Model

In this model, the association between the template molecule and the functional monomers is based on noncovalent interactions such as hydrogen bonds, Van der Waals forces or electrostatic interactions (Figure 3). Due to the wide commercial availability of monomers and analytes capable of giving rise to such stable interaction, this approach is the most currently used [6].

The polymerization process can be thermal or photochemical. The selection of any of these approaches depends on the template molecule as well as on the stability of its interactions with the functional monomers. If the template molecule is heat-sensitive, polymerization takes places at 4 °C using a UV-light source [7]. In the case of a photosensitive molecule, the temperature is used as a radical generator. If the molecule is both heat and light sensitive, the synthesis can be carried out avoiding radical initiation. This is the case for imprinted sol-gel polymerization that is initiated simply by an acid or base catalysis, eliminating the need for temperature or radiation that can damage the template molecule.

The basic components for the preparation of MIPs by a noncovalent imprinting approach are a functional monomer, cross-linker, initiator, and porogenic solvent [8].

##### Functional Monomer

The functional monomer is selected depending on the nature and features of the template molecule. For example, if the template has basic functional groups, the use of precursors with acidic moieties, such as acrylic acid or methacrylic acid, is preferred to facilitate the generation of ionic interactions or hydrogen bonds. On the other hand, if the template presents acid groups, the best choice would be to use weak bases, such as vinyl-pyridine. Finally, if the template is able to create stable complexes with certain ions, the use of functional chelating monomers such as vinylimidazole would be the best choice. The most common functional monomers used for the preparation of noncovalent MIPs are acrylic acid, methacrylic acid (MAA), p-vinylbenzoic acid, 2-vinylpyridine, 2-diethylaminoethyl methacrylate (DAEM), methacrylamide (MAM), and 2-hydroxyethyl methacrylate (HEMA) [9,10].

##### Cross-Linker

The addition of a cross-linker is necessary to create a three-dimensional network around the template. The most usual cross-linkers used for the preparation of noncovalent MIPs are divinylbenzene (DVB), ethylene glycol dimethacrylate (EGDMA), trimethylol propane trimethacrylate (TRIM), and pentaerythritoltriacrylate (PETRA) [11,12].

##### Initiator

The addition of a radical initiator is required in order to start the polymerization process by creating free radicals [13]. The most common initiators are azobisisobutyronitrile (AIBN), 2,2′-azobis(2,4-dimethyl)valeronitrile (ABDV), and benzoyl peroxide (BPO) [14].

##### Solvent

The solvent, or porogen, governs the strength of the interactions between the template molecule and the functional monomer(s) and influences the final morphology of the polymer [15]. The solvents with medium polarity, such as acetonitrile or chloroform, which increase the strength of the noncovalent interactions or those with a low dielectric constant, such as toluene or dichloromethane, are the most used for the synthesis of MIPs. Furthermore, the solvent also controls the superficial area and the pore size of the generated polymer, which can be critical in the recognition process, as a small pore size may hinder the diffusion of molecules into the binding cavities.

### 1.2. Physical Formats

Molecularly imprinted materials can be prepared in different physical formats, such as bulk, thin films, particles, or spheres, each of which has advantages and disadvantages depending on their final applications.

#### 1.2.1. Bulk Polymers

This type of polymerization was the first strategy applied for MIP synthesis, and it is still the most used due to the fact of its simplicity. The synthesis is based on a thermal or photochemical radical polymerization with the result of an insoluble polymer monolith, which is further crushed, ground, and sieved in order to obtain amorphous micrometric-sized particles [16].

The main disadvantages of this approach are the loss of material during sieving, the loss of binding sites during the crushing process, and the wide distribution of sizes and shapes of the particles.

#### 1.2.2. Thin Films

It is possible to prepare an imprinted polymer in the form of thin films onto a solid support (glass slides, electrodes, etc.) by different grafting techniques. This approach has several advantages, such as easier removal of the template after polymerization and fast equilibration with the analyte. Moreover, the binding sites will remain unaltered after template extraction, as no further polymer processing (grinding and sieving) is required before polymer use. On the other hand, the binding capacity of the resulting polymers is usually low due to the small amount of the polymer generated [17].

#### 1.2.3. Spherical–Nanowire Particles

Over the last years, the development of MIPs has focused on the synthesis of uniform-sized micro/nanoparticles with a narrow size distribution and improved mass transfer efficiency. Uniformly sized spherical particles can be prepared using various polymerization techniques, such as precipitation, suspension, or core–shell polymerization. The beads prepared by precipitation or emulsion polymerization sometimes do not offer easy access to their center, which hinders the washing and recognition processes [18].

In order to avoid this problem, core–shell particles can be prepared from silica, polystyrene, magnetic particles, or polymer particles, depending on the application [19]. This is a two-stage process in which a thin layer of polymer is grown on the seed bead surface that was previously activated. In general, silica is used as a support, because its functional groups allow for the anchoring of the polymer; thus, the binding sites are more accessible to the template molecule improving the mass transference. The silica is filled with the prepolymerization mixture occupying the holes dominated by the mesopores of its material. After the polymerization, the silica is dissolved with an appropriate solvent [20].

#### 1.2.4. Magnetic Molecularly Imprinted Polymers (mMIPs)

As mentioned previously, the spherical format allows for the synthesis of magnetic molecularly imprinted polymers (mMIPs) by growing an MIP layer on a magnetic nanoparticle. The core is usually formed by superparamagnetic nanoparticles, composed of Fe_2_O_3_ and Fe_3_O_4_, that can be easily attracted by an external magnetic field, which makes them easy to manipulate [21].

In recent years, magnetic nanoparticles have attracted great interest for their unique properties for the removal of toxic target compounds. Magnetic nanoparticles, especially iron oxide nanoparticles, represent a unique class of nanomaterials that have a very strong potential in biomedicine. Due to the fact of their good biocompatibility, stability in physiological environments, and size-dependent magnetic properties, they also play an important role in the development of diverse techniques, such as magnetic field controlled target delivery, bioseparations, and image contrast agents in magnetic resonance imaging (MRI) [22].

The use of magnetic particles together with solid-phase extraction sorbents is of particular interest for the clean-up and preconcentration of contaminants in complex food samples, as they facilitate sample manipulation improving the performance of the separation and washing steps. The application of mMIPs as sorbents for this aim will also allow for the selective extraction of the target analytes from the complex samples, which is of particular interest when impurities can interfere with quantification [23,24] (Figure 4).

### 1.3. MIPs Applications

MIPs are biomimetic nanostructures that can be used in a number of applications, including analytical separations, sensors, drug delivery systems, or catalysis [26].

#### 1.3.1. Synthesis and Catalysis

The MIPs synthesized using chiral or prochiral templates have binding sites that are asymmetric and similar to the domain-binding sites of enzymes but with more conformational restrictions. The spatial definition and the directionality of the chemical groups offer a high potential for the selective and asymmetric synthesis inside the cavities [27].

#### 1.3.2. Sensors

One of the most promising applications of MIPs consists of its use as selective recognition elements in sensors. The MIP is in contact with a transducer (optical, piezoelectric, amperometric, etc.), which transforms the chemical signal obtained in the course of the interaction with the analyte in an electrical signal that is easily quantified [28].

The imprinted materials allow for a high specificity of detection even in complex matrices at low concentrations of analyte and in extreme conditions (i.e., high temperature). Furthermore, the high physical and chemical stability of these polymers offers many advantages versus the use of the most frequently used biomolecules for the process of these devices [29].

#### 1.3.3. Solid-Phase Extraction (SPE)

Solid-phase extraction using molecularly imprinted polymers (MISPEs) is probably their most important field of application. This method requires a selective solid-phase to isolate the analyte of interest from a solution according to their physical and chemical properties. It is usually used for sample clean-up and preconcentration before chromatographic separation and/or quantitative target analysis [30,31].

#### 1.3.4. Analytical Separations

MIPs have been used as stationary phases in chromatography (HPLC), especially for the separation of isomers, or in capillary electrochromatography (CE). However, the most frequent application has been the characterization of the synthesized material using frontal and zonal chromatography [32].

In Spain, research in this area has focused mainly on the preparation of MIPs used for the selective recognition of target analytes and also for proteins’ purification. In addition, new polymerization methods to prepare these synthetic materials have also been described in the literature. The main research carried out on MIPs in recent years in Spain is described below.

## 2. Recognition and Determination of Target Analytes

### 2.1. Pesticides

Pesticides and herbicides are frequently present in natural and drinking waters, fruits, and vegetables due to the fact of their wide use in the agricultural industry. These compounds are considered hazardous for human health due to the fact of their toxicity. In fact, some of them have been classified by the Environmental Protection Agency as compounds likely to be carcinogenic to humans.

The determination of pesticides and herbicides is normally carried out by high-performance liquid chromatography or gas chromatography coupled to a detector (mass spectrometry, diode array detection, or atomic emission). In these methods, matrix components can interfere with the detection, making it difficult to reach the LODs established by regulations. Thus, sample treatment becomes a key step. In this sense, the research focuses on finding suitable sorbents to remove matrix interferences, improving the selectivity of the extraction process and reducing the use of organic solvent. Furthermore, these methods are laborious and time consuming, require expensive instrumentation, and do not allow for “in situ” and real-time monitoring. In this way, two sensing devices have been developed in Spain as promising alternatives to the former one [33,34].

Díaz-Álvarez et al. (2018) [33] reported the use of MIP microspheres, packed inside the hollow fiber, along with HPLC-UV detection as a selective and sensitive method for triazine determination in agricultural soil samples. The same authors (2022) [34] described the preparation of water-compatible MIPs for the recognition of several triazine herbicides directly in environmental waters and its determination by HPLC-UV detection. This MISPE procedure allows for the direct selective recognition of the target analyte, overcoming the conventional MIPs’ drawback of a lack of selective recognition in water samples.

Fresco-Cala et al. (2020) [35] described a monolith based on a molecularly imprinted polymer film immobilized on a carbon nanotube surface for the selective recognition of triazines herbicides in peppermint and tea samples. The selective monolith was prepared inside a stirring extraction unit, consisting of a polypropylene tube pierced with an iron ware. The quantification of the analytes was carried out using a gas chromatograph-mass spectrometer. Unlike MIPs immobilized on carbon nanotube surfaces, this approach provides a three-dimensional and macroscopic porous structure, making it easier to separate the nanoparticles from the sample once the extraction is complete.

Sanchez-Barragán et al. (2007) [36] described an MIP-based fluorescent flow-injection system (Figure 5) for the selective and sensitive determination of methyl-carbamate pesticide carbaryl in water based on the combination of an MIP for analyte recognition and luminescence measurements for its determination. This cost-effective method provides a selectivity and sensitivity towards the analyte as good as that offered by chromatography-based determination methods; it does not require sophisticated instrumentation, allows for “in situ” measurements, and the polymer can be used for up to 200 cycles, with high stability for more than 4 months.

Capoferri et al. (2018) [37] developed a highly sensitive and selective electrochromic sensor for the determination of chlorpyrifos, an organophosphate insecticide, in spiked water samples. This novel sensing device is based on the combination of an MIP, for the selective recognition of the analyte, and an electrochromic material of iridium oxide (IrOx), which exhibits the reversible change of its color in response to an externally applied potential. The pesticide quantification was carried out by direct visual detection and smartphone imaging, as shown in Figure 6.

This article presents a portable, fast, low-cost, and easy-to-use sensor for the real-time and in situ determination of insecticides that takes advantage of the great potential of smartphones.

### 2.2. Mycotoxins

Mycotoxins are toxic secondary metabolites produced by certain types of fungi that can be found in a variety of crops and food, such as cereals, nuts, spices, fruits, dried food, and coffee beans. Mycotoxins appear as a result of a fungi infection of crops both before and after harvest, during storage, or in the food itself under warm and humid conditions. These toxic compounds are capable of causing adverse health effects in humans and animals, such as reproductive and infertility problems, cancers, and immune deficiency. Therefore, the development of accurate fast methods for the determination of these mycotoxins is of the utmost importance to protect humans and animals from their toxic effects.

The determination of mycotoxins is normally carried out by high-performance liquid chromatography (HPLC), with UV fluorescence or mass spectrometry, or by gas chromatography-mass spectrometry. These methods require sample extraction and clean-up procedures to remove matrix components that can interfere with the detection. Liquid–liquid extraction is time consuming and involves the high consumption of solvents. Solid-phase extraction (SPE) sorbents are poorly selective. Enzyme-linked immunosorbent assays (ELISAS) and immunosensors are highly selective but are expensive, have chemically and thermally low stability, and are not easy to prepare. MIPs can be a superior alternative to existing methods since they lack the aforementioned drawbacks.

Urraca et al. (2006) [38] prepared MIPs in bulk format for the selective recognition of zearalenone (ZON) in contaminated food samples, using a ZON analogue (cyclodecyl 2,4-dihydroxybenzoate, CDHB) as the template instead of a natural toxin. The MIPs synthesized using 1-allylpiperazine (1-ALPP) as the functional monomer, TRIM as the cross-linker, AIBN as the initiator, and acetonitrile as the solvent showed the largest capacity and binding affinity to ZON. The target analyte determination was carried out by HPLC coupled to a UV fluorescence detector.

Rico-Yuste et al. (2021) [39] reported the development of a sensitive and selective luminescent sensor based on a Eu(III) ion-doped MIP for the selective and fast determination of tenuazonic acid (TeA) in rice extracts at the few μg mL^−1^ level. The MIPs were prepared in a porous microsphere format using two different functional monomers, diethyl allylmalonate (DEAM) and allyl acetoacetate (AACA), TeA as the template molecule, EGDMA as the cross-linker, and ABDV as the initiator, generating Eu(III)-imprinted polymers. The determination of the analyte is based on the enhancement of the sensor emission at 615 nm, as a result of TeA binding to the lanthanide. Although the LOD of the optosensor was far from the one provided by mass spectrometry, the novel luminescent sensor is a promising alternative, as it is rapid, affordable, easy to use, and allows for the simultaneous analysis of many samples, reducing the analysis time compared to existing methods.

Quílez-Albuquerque et al. (2021) [40] described an MIP-based luminescent sensor for TeA detection by both luminescence intensity and lifetime measurements. The MIP core–shell nanoparticles were prepared using a ruthenium(II)-bipyridyl complex ([Ru(dab)_2_(bim)]^2+^) as the functional monomer, MAM as the diluent monomer, TeA as the template molecule, EGDMA as the cross-linker, and ABDV as the initiator (Figure 7). The determination of the analyte is based on the change of the probe emission features (luminescence intensity and lifetime) as a result of TeA binding to the complex. The novel optosensor provides good sensitivity and selectivity towards the analyte and is suitable for real-time measurements due to the fast response displayed.

### 2.3. Antibiotics

Sulfonamides and fluoroquinolones (FQs) are broad-spectrum antibiotics widely used for the treatment of a wide variety of infections in humans as well as in livestock intended for human consumption and in aquaculture farming. The intensive use of these antibiotics leads to the spread of antimicrobial resistance that is becoming a worldwide serious concern. To prevent their negative effects in humans, animals, and the environment, strict regulations for antibiotics usage must be made, and new analytical methods to determine their presence in the environment and food must be developed.

Several analytical methods have been proposed for the analysis of these antibiotics, such as chemiluminescence, UV-Visible, spectrophotometry, HPLC-MS/MS, and electrochemical methods. The determination of these analytes cannot be carried out directly; it requires a sample pretreatment and/or extraction techniques. For this task, selective sorbents are required. MIPs have proven to be excellent sorbents in SPE due to the fact of their selectivity towards the analyte, reusability, stability, and shorter preparation times.

Guzmán-Vázquez de Prada et al. (2005) [41] reported a method for the determination of sulfamethazine (SMZ) in milk based on the use of an MIP for the selective solid-phase extraction of the antibiotic coupled to a sensitive voltammetric detection. The MIP was prepared using SMZ as template molecule, MAA as the functional monomer, EGDMA as the cross-linker, and acetonitrile as the solvent. The method described in this work allows for the quantification of SMZ at the concentration levels required by legislation in complex samples such as milk.

Barahona et al. (2019) [42] described the preparation of MIP immobilized in the pores of polypropylene hollow fibers (HFs) as selective microextraction methodology for the detection of fluoroquinolones (FQs) in water and urine samples at the few μg L^−1^ level. The MIP-HFs were prepared using enrofloxacin (ENRO) as the template molecule, MAA as the functional monomer, EGDMA as the cross-linker, AIBN as the initiator, and toluene as the porogen (Figure 8).

Real samples were analyzed by HPLC coupled to a mass spectrometer (LC-MS/MS). This methodology improves the LODs previously obtained, provides selectivity towards the analyte, reduces the use of organic solvents, and is suitable for real sample analysis. However, these methods are laborious, require expensive instrumentation, and do not allow for “in situ” and real-time monitoring.

Baeza et al. (2022) [31] recently reported the development of a highly sensitive and selective online solid-phase extraction methodology based on a selective MIP and HPLC-FLD detection for FQ determination in river water samples (Figure 9). The MIPs were prepared using enoxacin (ENOX) as the template molecule, MAA and TFMAA as the functional monomers, EGDMA as the cross-linker, ABDV as the initiator, and acetonitrile as the solvent. The described method was highly sensitive and useful for the trace determination of FQs in real samples, achieving LODs in the low ng L^−1^ level.

Spiramycin is a macrolide antibiotic used to treat infections in animals. The administration of these antibiotics may leave residues in food animal origin, such as milk, egg, and meat, causing diseases or disorders in consumers. European Union regulation authorities have established the maximum limits of these macrolides residues in milk. The existing analytical methods are not sensitive enough. Therefore, new analytical methods need to be developed.

García-Mayor et al. (2017) [43] described the preparation and evaluation of a series of MIPs as sorbents for the extraction and preconcentration of spiramycin (SPI) from aqueous and sheep milk samples. The analyte detection was carried out by HPLC with UV diode-array detection. The MIPs were prepared by bulk polymerization using SPI as the template molecule, MAA as the functional monomer, EGDMA as the cross-linker, AIBN as the initiator, and acetonitrile as the solvent. The developed method provided an LOD lower than the one established by current regulations, allowing for a rapid, sensitive, and cost-effective analysis of SPI in milk samples.

### 2.4. Illicit Drugs

The continued growth of the new psychoactive substances (NPS) market has become a major concern. The range of drugs available on the market has probably never been wider. These NPS cause similar or even greater effects than that provoked by similar substances of natural origin. However, the structural differences between synthetic and natural drugs allow these new analogs to avoid detection during toxicological–forensic analysis. Therefore, new screening assays are needed. An analytical methodology based on liquid chromatography and liquid chromatography coupled to mass spectrometry detection requires sample pretreatments to remove matrix interferences. MIPs have proven to be excellent sorbents to selectively extract target analytes, providing selectivity for the method, especially when the samples are complex.

Sánchez-González et al. (2018) [44] described the synthesis of MIPs as new selective adsorbents for the micro-solid-phase extraction (μ-SPE) of twenty synthetic cannabinoids in urine samples. The MIPs were prepared using synthetic cannabinoids as the template molecules, EGDMA as the functional monomer, divinylbencene-80 (DVB) as the cross-linker, AIBN as the initiator, and a mixture of acetonitrile and toluene as the solvent. The developed approach consisted of a porous polypropylene membrane containing MIP particles, allowing for a fast and cost-effective extraction, clean-up, and preconcentration (Figure 10). The analyte detection was carried out by HPLC-MS/MS.

The same research group [45] applied the same μ-SPE device for the determination of synthetic cathinones, which are synthetic drugs that have stimulant effects similar to cocaine and methamphetamine. The MIPs were prepared as previously described, using ethylone and 3-methylmethcathinone as the templates. The cathinones determination was carried out by HPLC-MS/MS, obtaining the same good results.

Sorribes-Soriano et al. (2019) [46] described a new extraction approach based on rotating MIP polytetrafluoroethylene (PTFE) disks. The MIP synthesis was carried out by in situ polymerization onto the surface of the PTFE disks, which were previously modified. The extraction device can be magnetically stirred by placing two ring magnets on the sides of the MIP disk (Figure 11). The novel system was used for the analysis of ecgonine methyl ester (EME), a cocaine metabolite, in water samples. The analyte determination was performed by ultra-high-performance liquid chromatography coupled to mass spectrometry (UHPLC-MS/MS) and ion mobility spectrometry (IMS), the former being more sensitive and IMS being quicker. The new method provides a simple, selective, and effective extraction of the analyte along with a sensitive detection yielding limits of detection of 13 ng L^−1^ (UHPLC-MS/MS) and 75 ng L^−1^ (IMS).

Díaz-Liñán et al. (2021) [47] reported the use of an MIP paper directly coupled to mass spectrometry via direct infusion electrospray ionization for the determination of cocaine and methamphetamine in saliva samples. The MIP paper was prepared by immersing the paper into a polymeric solution, using cocaine and methamphetamine as templates. The resulting analytical device is highly sensitive and selective, does not require invasive sampling and specialized personnel, and is fast, as the MIP paper can be directly coupled to mass spectrometry.

### 2.5. Explosives

The presence of explosive-related compounds and their degradation products in water has become a major concern. For instance, 2,4,6-trinitrotoluene (TNT) has been classified as a probable human carcinogen. According to the United States Environmental Protection Agency, TNT in drinking water poses a considerable risk at concentrations above 0.44 μmol L^−1^ [48]. Therefore, reliable monitoring methods for these analytes are needed. Several analytical methods have been described, such as HPLC or GC coupled to mass spectrometry. These methods are laborious, require expensive instrumentation, and require sample pretreatment. Electrochemical sensing is suitable for the determination of nitroaromatic species, due to the fact of their inherent electroactivity, and it is cost effective, allows for portable equipment, and does not need sample pretreatment. Hence, electrochemical sensors can provide fast, “in situ”, and reliable measurements.

Herrera-Chacon et al. (2021) [49] reported a voltammetric sensor based on an MIP for the determination of TNT. The MIP synthesis was performed by thermal precipitation polymerization using 2,4-dinitrophenol (DNP) as the dummy template, MAA as the functional monomer, EGDMA as the cross-linker, AIVN as the initiator, and ethanol as the solvent. The polymer microbeads were integrated into graphite epoxy composite electrodes, via sol-gel immobilization, and used as a voltammetric sensor. The sensor was rapid, low cost, and sensitive enough (LOD of 0.29 μmol L^−1^) to detect below the concentrations at which TNT is considered a risk.

Explosive detection has also become important in military operations, homeland security, and environmental safety. The detection of some explosives such as TNT or 1,3,5-trinitroperhydro-1,3,5-triazine (RDX) in a gas state is pretty difficult due to the fact of their low vapor pressure at room temperature. Existing analytical techniques, such as HPLC-MS/MS, surface-enhanced Raman spectroscopy, and X-ray diffraction, are sensitive and selective, but most of them are expensive and require bulky and sophisticated instrumentation, which limits “in situ” measurements. Hence, fast and portable detection devices are needed in these contexts.

Aznar-Gadea et al. (2022) [50] developed two luminescent gas sensors based on a nanocomposite of CsPbBr_3_ nanocrystals (NCs) embedded in an MIP for the detection of 3-nitrotoluene (3-NT) in gas samples. The MIP sensors were prepared using 3-NT and nitromethane (NM) as the template molecules, using a fast and low-cost process. A solution of CsPbBr_3_ NCs, polycaprolactone (PCL), and template molecules was spin-coated onto a substrate to generate a thin film and baked (see Figure 12). The resulting sensors were sensitive and selective, and they showed a fast response time (below 5 s), with the NMMIP sensor being the one that exhibited the highest response to the analyte. The latter provided fast detection (2–3 s) of3-NT, with a LOD of 0.218 mg mL^−1^.

### 2.6. Biological Analytes

Montoro-Leal et al. (2022) [51] reported an electrochemical sensor based on a modified screen-printed carbon electrode (SPCE) for the electrochemical sensing of malondialdehyde (MAD) in a serum sample. This molecule is an important biomarker of oxidative stress used in medical diagnosis and is involved in serious human diseases, such as diabetes, heart disorders, and cancer. The new nanocomposite material used for the detection of MAD was prepared by coating the magnetic graphene oxide (MGO) with a molecularly imprinted polypyrrole (MIPy). For the MIPy synthesis, malondialdehyde derivatized with diaminonaphthalene (MDA-DAN) was used as the template molecule. The new sensing system was used to analyze three chicken serum samples, proving its suitability for MDA quantification with high sensitivity and precision. This electrochemical sensor is a promising alternative to the existing analytical methods, as it is low cost and can be easily miniaturized.

The same research group (2022) [52] developed a molecularly imprinted electrochemical sensor based on a nanocomposite for the routine monitoring of lysozyme (LYS) in chicken eggs and in a commercial drug. LYS is a potential biomarker for the diagnosis of leukemia and other diseases. It is used as an ingredient in pharmacology due to the fact of its antibacterial, analgesic, and anti-inflammatory effects, as well as in the food industry as preservative. The nanocomposite was prepared by the polymerization of pyrrole (PPy) in the presence of multifunctional graphene oxide/iron oxide composite (GO@Fe_3_O_4_) and was electrodeposited on top of a gold microelectrode array (Figure 13).

The novel sensor was highly sensitive and selective, provided a low LOD (0.009 pg mL^−1^), and was easy to use, reliable, cost-effective, and reusable up to nine times. Therefore, this sensing device is a good alternative to existing analytical methods (HPLC, MS, ELISA, and electrophoresis), as in most cases they require expensive equipment, sample pretreatments, or long analysis times.

### 2.7. Other Analytes

Masqué et al. (2000) [53] described the synthesis and use of an MIP as a selective SPE sorbent to selectively extract the environmental pollutant 4-nitrophenol (4-NP) from river water samples. The application of the MISPE to environmental water samples confirmed its usefulness for sample clean-up and preconcentration before chromatographic separation and quantitative target analysis.

Herrera-Chacon et al. (2018) [54] proposed a new bio-electronic tongue (BioET) analysis system based on the use of MIPs as a recognition element and in differential pulse voltammetry (DPV) for the detection of volatile phenols during wine production. Volatile phenolic compounds, such as 4-ethylphenol (4-EP) and 4-ethylguaiacal (4-EG), are produced by Brettanomyces yeast during wine fermentation stages. Their detection during wine production is of great importance, as these compounds can change the flavors and aromas of the wine, causing losses in the beverage industry. The human threshold is approximately 0.5 μg mL^−1^ for 4-EP. The MIP was synthesized using 4-EP and 4-EG as the template molecules, DVB as the functional monomer, EGDMA as the cross-linker, AIVN as the initiator, and ethanol as the solvent. The MIP particles were integrated onto an electrochemical sensor surface using the sol-gel technique in the presence of graphite as the conducting material. The resulting biosensor electrodes were applied for the quantification of 4-EP and 4-EG mixtures with LODs of 1.3 and 2.4 μg mL^−1^, respectively.

Ruiz-Córdova et al. (2018) [55] described a proof-of-concept electrochemical sensor based on the use of magnetic MIP particles (mag-MIPs) and a magneto-sensor that acts as the working electrode for the determination of 1-chloro-2,4-dinitrobenzene (CDNB) using differential pulse voltammetry (DPV). This work’s novelty consists of the use of a magneto-sensor that concentrates the mag-MIPs on its surface, allowing a direct voltammetric analysis of CDNB (Figure 14).

Alnaimat et al. (2020) [56] reported a sensitive and selective method based on the use of MIP for solid-phase extraction of phthalates from tea samples and their analysis using HPLC coupled to mass spectrometry. Phthalates are phthalic acid esters used as plasticizer in many industrial products. These compounds can be liberated from plastic containers into food and drink, posing risks to human health. Phthalates are used to fabricate filter bags for the packaging of tealeaves. Hence, these compounds are found in tea infusions. The reported method was successfully applied for the analysis of butyl benzyl phthalate (BBP), diethyl phthalate (DEP), dibutyl phthalate (DBP), and dimethyl phthalate (DMP) in several tea samples.

## 3. Contaminant Removal

Contaminants of emerging concerns (CECs) are chemical compounds that are persistent towards conventional wastewater treatments and potentially harmful to the environment and humans. Diclofenac (DCF) and indomethacin (IDM) are anti-inflammatory drugs frequently detected in the aquatic environment, including river and surface waters. Existing water treatments are not suitable for their removal. Hence, new methods need to be developed.

Samah et al. [57,58] described the preparation of MIPs for the removal of DCF and IDM from water. MIPs were prepared by bulk polymerization using DCF or IDM as the template molecule, 1-allylthiourea as the functional monomer, EGDMA as the cross-linker, AIBN as the initiator, and acetonitrile as the solvent. The developed method provides a fast and less laborious method for the selective and highly effective removal of DCF and IDM from wastewater.

## 4. Proteins Purification

Bioaffinity chromatography is one of the most powerful techniques for the purification of biological molecules. However, the high price and limited reusability of bioaffinity columns prevent their widespread use in biotechnology application. MIPs can overcome some of these shortcomings for such applications due to the fact of their long-term stability, cost, and reusability. The FLAG tag (DYKDDDDK) is a short peptide used for the purification of recombinant peptides. Purification requires the use of anti-FLAG antibody resins that are costly and nonreusable. Gómez-Arribas et al. (2019) [59] reported an alternative strategy based on the use of MIPs synthesized using the tetrapeptide DYKD as the template for the purification of FLAG-derived recombinant proteins. A combinatorial MIP library was prepared (Figure 15). The polymer with the best specific affinity towards FLAG and the peptide DYKD was the one prepared using DYKD as the template, EAMA as the functional monomer, EGDMA as the cross-linker, ABDV as the initiator, and dimethylformamide (DMF) as the porogen. The resulting MIP was used for the purification of mCherry proteins tagged with either FLAG or DYKD epitopes. Both mCherry variants were highly efficiently purified.

Although good results were obtained with the MIP, the recognition of the FLAG tag showed nonspecific binding when compared with the nonimprinted polymer (NIP). In an attempt to suppress this effect, the same research group (2020) [60] reported the synthesis of MIPs selective to the FLAG by a hierarchical imprinting strategy. For the MIP preparation the template molecule, 5-amino acid peptide DYKDC, was covalently immobilized on the surface of porous silica beads, previously modified with different aminosilanes, packed in SPE cartridges and subsequently filled with the prepolymerization mixture with EAMA as the functional monomer, EGDMA as a the cross-linker, 2,2-azobis(2,4-dimethylvaleronitrile) as the initiator, and DMSO as the porogen (Figure 16).After the template and silica removal, the resulting MIPs were evaluated as sorbents for the solid-phase extraction of the FLAG peptide, showing unprecedented selectivity towards the peptide. Hence, the optimized hierarchical MIP can be considered a promising material for the FLAG-tagged protein purification.

## 5. New Polymerization Methods

Conventional methods for MIP preparation, such as bulk polymerization and precipitation polymerization, require stringent control of the experimental conditions, making them complex. Hence, simpler, easier, and faster methods are needed. Díaz-Liñán et al. (2019) [61] suggested a polymerization-free method for the preparation of an MIP based on the dissolution of a polymer (nylon-6) in an appropriate solvent (formic acid), and the later addition of the porogen and template molecule. The subsequent immobilization of the MIP on a filter paper and removal of the template provides an MIP paper-based analytical device that is coupled to a fluorometer for the selective determination of quinine (Figure 17). It is an easy, rapid, and cost-effective method that can be used to prepare paper-based analytical devices that can be coupled to different detectors, developing detection methods for different analytes.

In recent years, magnetic MIP nanoparticles (MNP-MIPs) have attracted great interest as sorbents for the selective extraction of analytes in complex samples. The core is usually formed by superparamagnetic nanoparticles composed of Fe_2_O_3_ and Fe_3_O_4_. The use of magnetic fields to remove MNP-MIPs from solution is much faster and easier when compared to filtration or centrifugation. The existing methodologies for these magnetic MIP nanoparticles, such as free radical polymerization and controlled radical polymerization, have their own drawbacks. The first method does not provide control over the size of the macromolecule. Although the second method leads to a more controlled polymer, it is slower, as it involves several steps, and it requires the use of special reagents. Urraca et al. (2018) [62] presented a simpler and faster approach to the preparation of magnetic MIP nanoparticles by using alternating magnetic fields to trigger the polymerization reaction. Magnetic nanoparticles were selectively heated by alternating magnetic fields, allowing the formation of a thin MIP layer onto their surface. The MIPs were synthesized using rhodamine 123 (R123) as the template molecule, MAA as the functional monomer, and TRIM as the cross-linker. This polymerization method can also be applied to deposit uniform polymer coatings onto other magnetic nanostructures such as nanowires (NWs).

Lamaoui et al. (2019) [63] described the synthesis of magnetic MIPs using a high-power ultrasound probe. The method was easier and faster than conventional ones. It was found that the effect of the solvent was critical, affecting the reactivity and the product yield. Magnetic MIPs were successfully synthesized using ultrasonic dissipated power solvents, such as DMSO, DMF, and ethanol, in less than 10 min. These MIPs were successfully applied as adsorbent materials in SPE coupled to a colorimetric detector to selectively determine sulfamethoxazole in water samples.

When it comes to electrochemical sensors based on MIPs, the integration of the recognition element on the transduction system is critical. Acrylic or vinyl MIP layers are electrically insulating and avoid the proper functioning of the electrode transducer. This shortcoming can be solved with thin MIP coatings (nm scale) or with the combination of MIP coatings and conducting materials, such as gold o nickel nanoparticles. García-Mutio et al. (2018) [64] proposed a new method for the controlled coating of gold microelectrodes with MIP thin films based on living polymerization mediated by a thiol iniferter. Once the iniferter is immobilized on gold, the synthesis of the polymer takes place at the surface of gold microelectrode. The thiol iniferter is separated into an active and a dormant species, controlling the process (Figure 18). It was found that irradiation time was critical to obtain thin coatings, avoiding electrode insulation. The resulting MIP-coated microsensor was evaluated for the determination of 4-ethylphenol, showing a linear electrochemical response and good sensitivity and selectivity towards the target analyte.

Pérez-Puyana et al. (2021) [65] reported the preparation of MIPs directly from polymers instead of using monomers as the starting materials, as structures for biological recognition. Poly(ε-caprolactone) (PCL) was selected as a synthetic polymer and three different proteins (gelatin, collagen, and elastin) as natural polymers. Molecular imprinting products were prepared by a two-stage process. First, membranes were obtained via electrospinning using mixtures of PCL and protein. Then, in the second stage, the template was removed by a solvent extraction step. Different PCL-protein scaffolds were prepared via electrospinning, providing an alternative method for the fabrication of structures with specific biological recognition sites. Their applicability has still to be evaluated.

Table 1 shows a summary of the applications of MIPs in Spain in recent years. They were classified according to the type of target analyte (pesticides, mycotoxins, antibiotics, Illicit drugs, explosives, biological analytes, or other analytes) and the type of function it has (sensor or MISPE). The composition of the MIP (template molecule, functional monomers, and cross-linkers), the method of polymerization used in each work (type of polymerization), the medium in which the polymer measurement parameters were optimized (means of measurement), the measurement system used in each case (validation methods), the real samples analyzed with the MIP, and the limit of detection (LOD) and quantification (LOQ) values obtained in the analysis of these real samples were also specified.

## 6. Conclusions and Future Outlook

Molecularly imprinted polymer technology is a promising tool in the development of sensing and biosensing devices, as it provides high selectivity and sensitivity towards the analyte and shows long-term stability and resistance in organic media, with the reusability being much higher than for biological materials. Moreover, MIPs are easy to prepare in a wide range of formats and are cost effective. As it has been described in this review, these materials have already been applied as selective sorbents for the extraction and/or determination of a variety of target analytes. Nevertheless, further research must be conducted to introduce MIP-based analytical devices on the market. One of the main challenges is the integration of MIP particles in transducers and assay formats suitable for commercial selling. New methods are trying to make the integration process feasible and simple. Lastly, the challenges are the large-scale production of MIPs with a homogeneous size and shape, as well as similar properties in the synthesis between different batches and the imprinting of bigger biological molecules such as proteins, where the recognition can be extrapolated to other systems, such as cells, viruses, and bacteria.

## Figures and Tables

**Figure 1 ijms-24-01915-f001:**
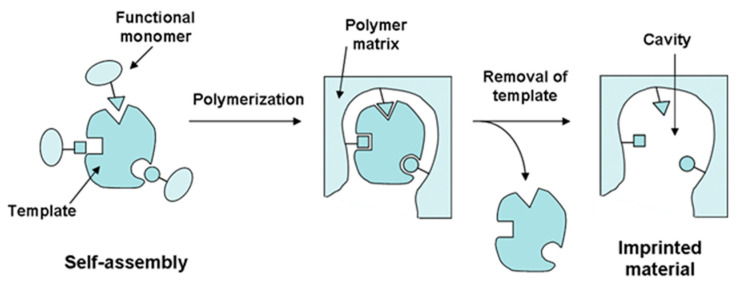
Schematic representation of the molecular imprinting process.

**Figure 2 ijms-24-01915-f002:**
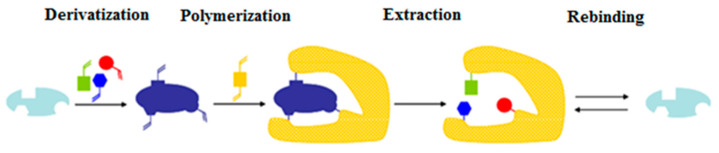
Schematic representation of the covalent imprinting approach.

**Figure 3 ijms-24-01915-f003:**
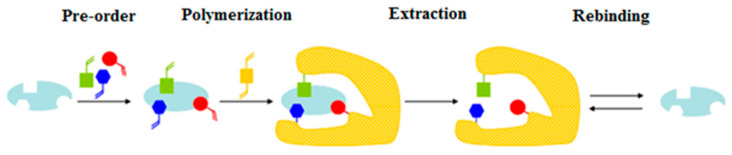
Schematic representation of the noncovalent imprinting approach.

**Figure 4 ijms-24-01915-f004:**
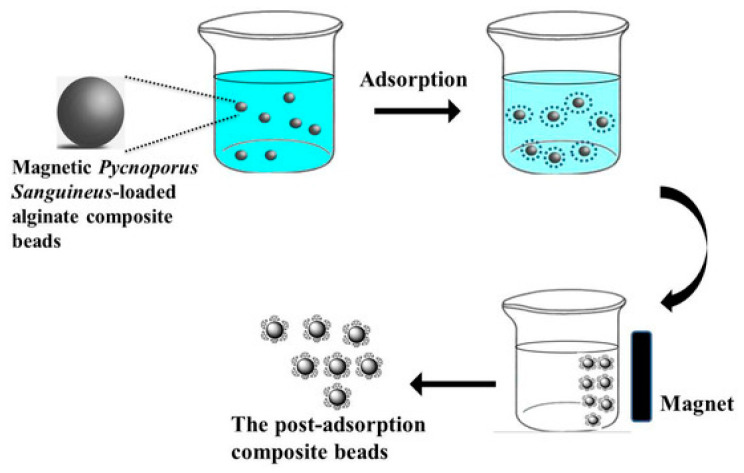
Schematic representation of the magnetic separation process. Reprinted from [25], with permission from Open Access MDPI, 2014.

**Figure 5 ijms-24-01915-f005:**
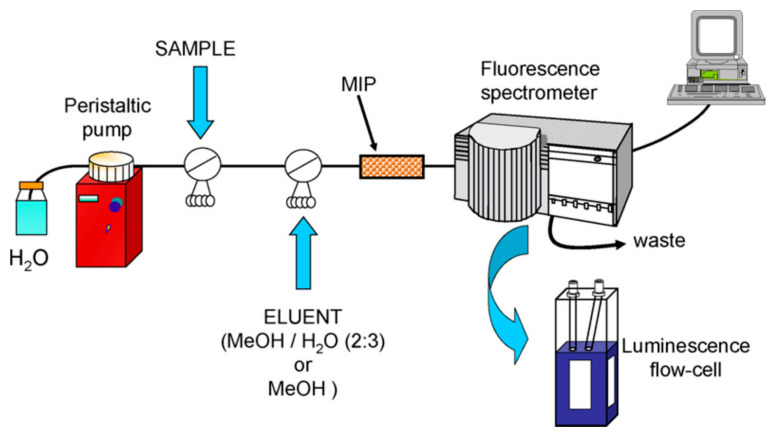
Optosensing manifold for the luminescent determination of the pesticide carbaryl. Reprinted from [36], with permission from Elsevier, 2007.

**Figure 6 ijms-24-01915-f006:**
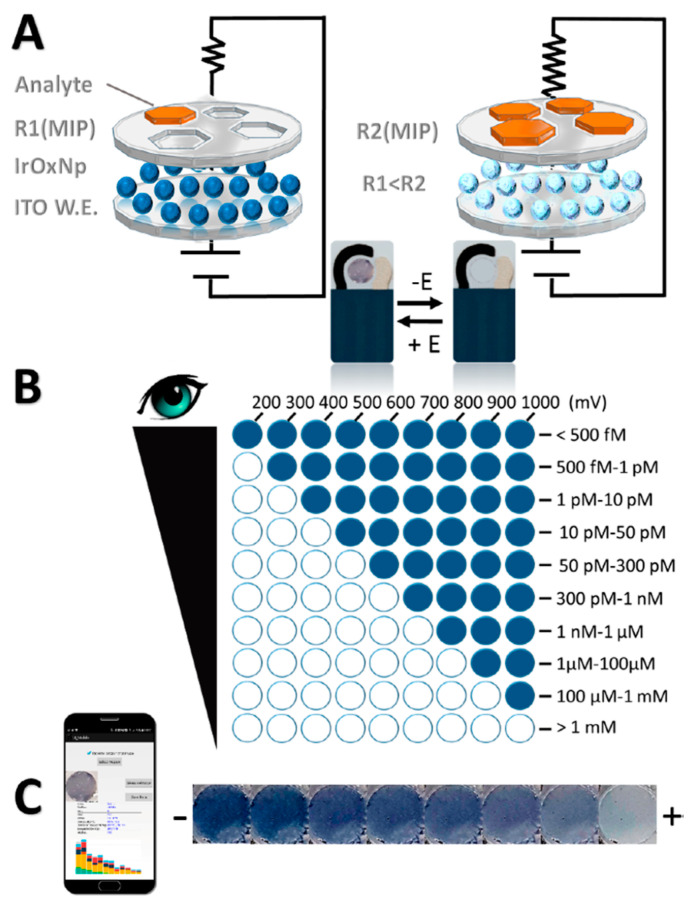
(**A**) Scheme of the MIP/IrOx NPs—ITO SPEs structure, visual IrOx NPs color change (from blue—black to transparent), and working principle of the proposed sensor with different amounts of analyte. (**B**) Visual detection after 10 s of the application of different oxidation potentials and concentration ranges detected based on the number of colored electrodes. (**C**) Change of the IrOxNPs’ color intensity at a fixed time and potential vs. increasing amounts of the analyte (smartphone-based detection). Reprinted from [37], with permission from American Chemical Society, 2018.

**Figure 7 ijms-24-01915-f007:**
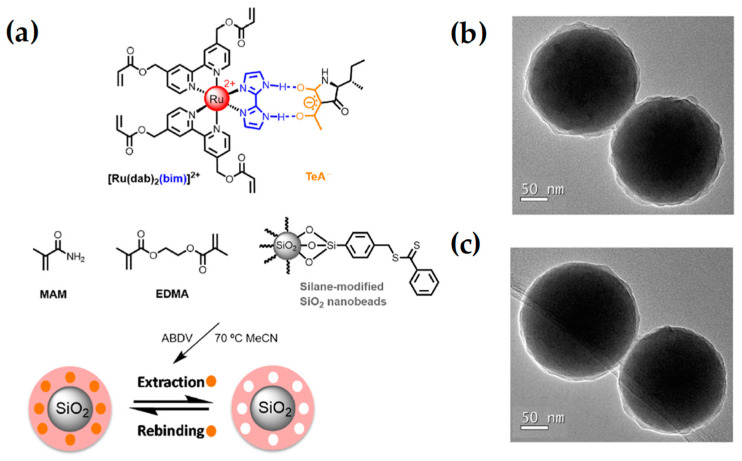
(**a**) Scheme of the preparation of the luminescent SiO_2_@Ru-MIP core–shell nanoparticles for TeA-recognition. The RAFT polymerization was initiated with ABDV (2,2′-azobis(2,4-dimethyl)valeronitrile) and heat. (**b**,**c**) TEM images of the obtained MIP and NIP core–shell nanoparticles, respectively. The orange circles represent tenuazonic acid (TeA) molecules. Reprinted from [40], with permission from Elsevier, 2021.

**Figure 8 ijms-24-01915-f008:**
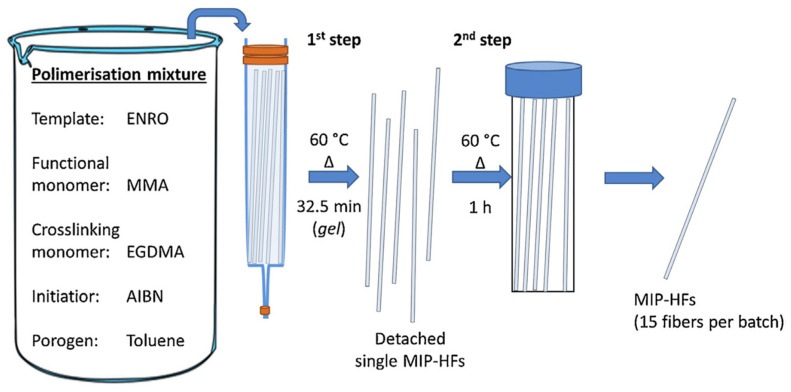
Scheme of the two-step polymerization procedure for the preparation of MIP-HFs. Reprinted from [42], with permission from Elsevier, 2019.

**Figure 9 ijms-24-01915-f009:**
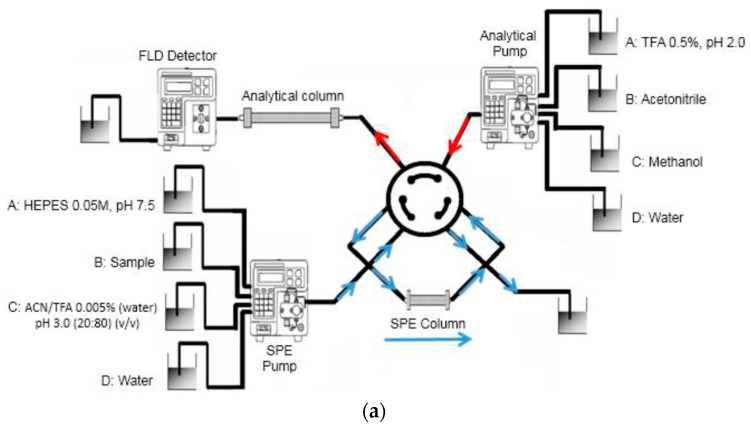
Scheme of the online MISPE-HPLC setup: (**a**) valve position A—sample loading; (**b**) valve position B—sample analysis. Reprinted from [31], with permission from Open Access MDPI, 2022.

**Figure 10 ijms-24-01915-f010:**
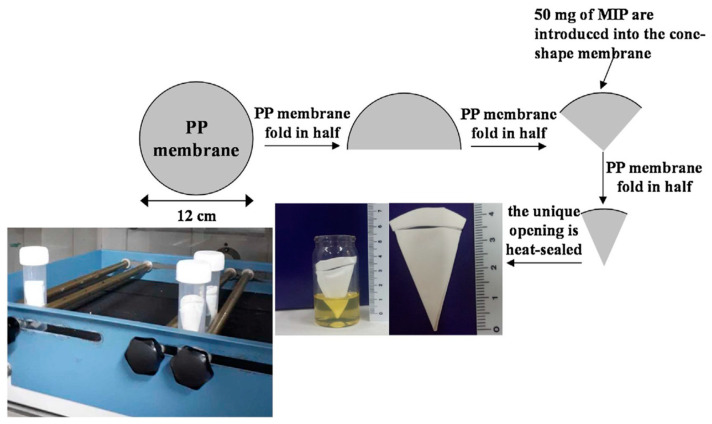
Schematic of the preparation of the MIP-μ-SPE device. Reprinted from [44], with permission from Elsevier, 2018.

**Figure 11 ijms-24-01915-f011:**
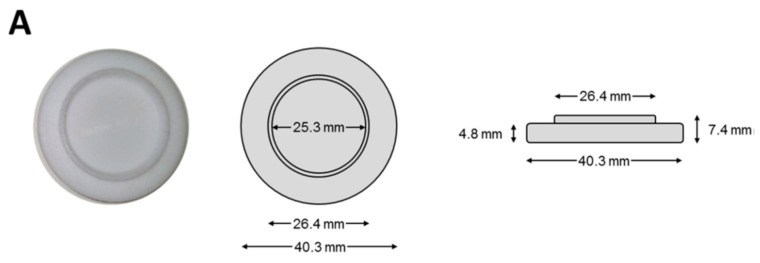
(**A**) Scheme and dimensions of the employed polytetrafluoroethylene (PTFE) disks; (**B**) polymerization assembling; (**C**) sampling assembling. Reprinted from [46], with permission from Elsevier, 2019.

**Figure 12 ijms-24-01915-f012:**
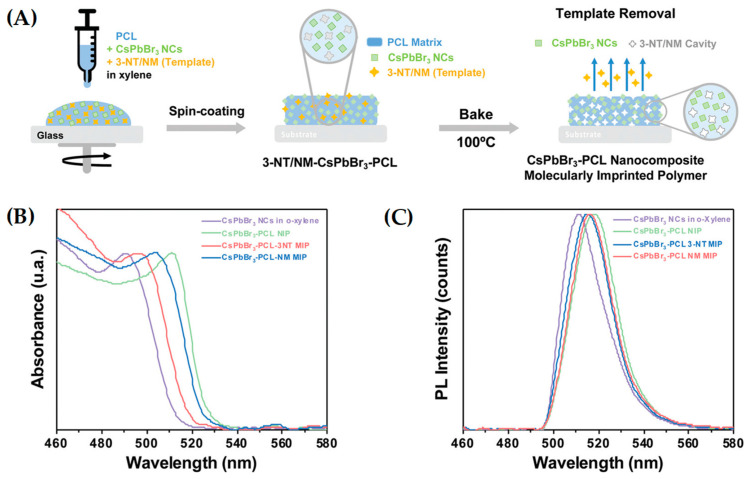
(**A**) Steps in the fabrication of the CsPbBr_3_–PCL 3-NT or NM MIP sensor; (**B**) UV-vis absorbance spectra and (**C**) emission spectra of CsPbBr_3_ NCs in o-xylene (purple line) and embedded in the PCL polymer matrix without (green line) and with a 3-NT (blue line) or NM (red line) template. Reprinted from [50], with permission from Royal Society of Chemistry, 2022 (https://pubs.rsc.org/en/content/articlehtml/2022/tc/d1tc05169e (accessed on 16 December 2022)).

**Figure 13 ijms-24-01915-f013:**
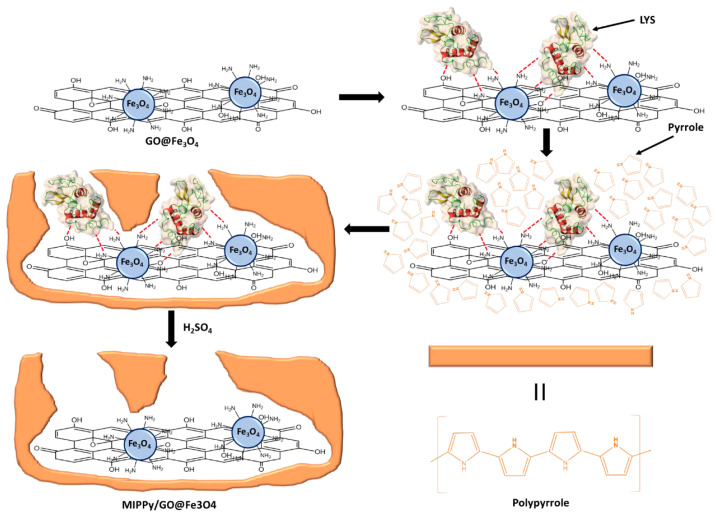
Synthesis process of MIPPy/GO@Fe_3_O_4_. Reprinted from [52], with permission from Open Access MDPI, 2022.

**Figure 14 ijms-24-01915-f014:**
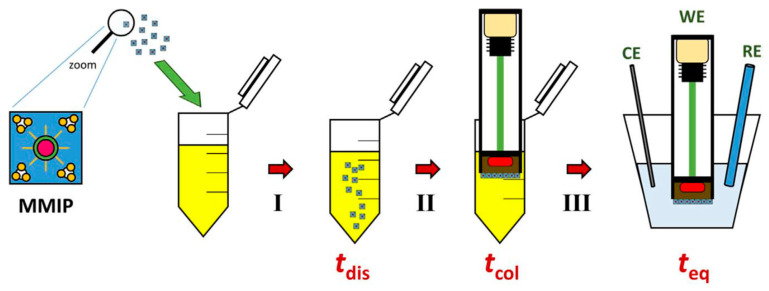
Schematic of the analytical procedure after placing the Mag-MIPs: (I) dispersion of the mag-MIPs (dispersion in a 3 mg mL^−1^ glycine buffer solution, at a pH of 2.0, 0.1 mol L^−1^, Tween 20, 0.05%, and NaCl 0.15 mol L^−1^); (II) magnetic capture by the magneto-sensor; (III) analyte binding to the mag-MIPs (in a 20 mL solution: 9 mL of sample, 9 mL of phosphate buffer solution, and 2 mL of ethanol). Reprinted from [55], with permission from Elsevier, 2018.

**Figure 15 ijms-24-01915-f015:**
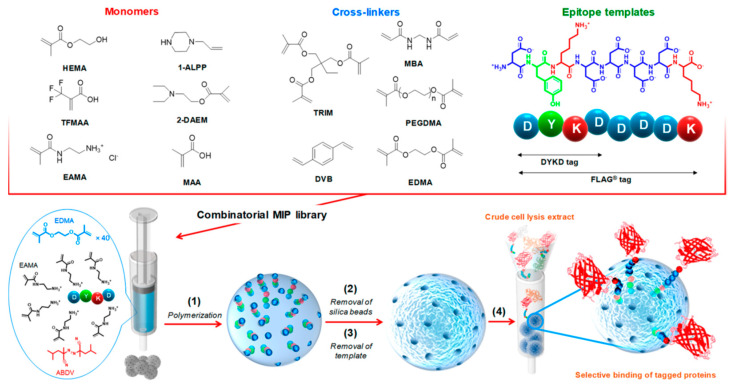
Workflow of the production of DYKD and FLAG tag-imprinted resins and purification of tagged mCherry-FLAG by MISPE. Reprinted from [59], with permission from American Chemical Society, 2019.

**Figure 16 ijms-24-01915-f016:**
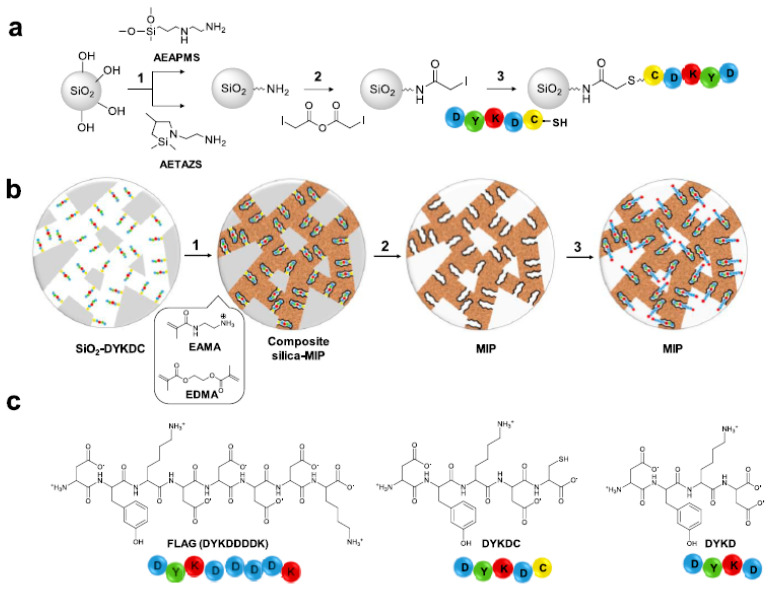
(**a**) Derivatization of silica particles: (1) silanization of rehydroxylated silica beads by using two types of silane agents: AEAPMS (3-(2-aminoethylamino)propyldimethoxymethylsilane) and AETAZS (N-(2-aminoethyl)-2,2,4-trimethyl-1-aza-2-silacyclopentane); (2) iodoacetylation of amino-modified silica particles; (3) peptide DYKDC coupling through the thiol group of the C-terminus cysteine. (**b**) Hierarchically imprinted polymer synthesis: (1) polymerization using EAMA (N-(2-aminoethyl)methacrylamide hydrochloride) as a functional monomer and ethylene glycol dimethacrylate (EGDMA) as a cross-linker; (2) template and silica removal; (3) FLAG recognition by epitope-imprinted cavities. (**c**) Chemical structures of FLAG (DYKDDDDK), DYKDC, and DYKD peptides, where D = aspartic acid, Y = tyrosine, K = lysine, and C = cysteine. Reprinted from [60], with permission from American Chemical Society, 2020.

**Figure 17 ijms-24-01915-f017:**
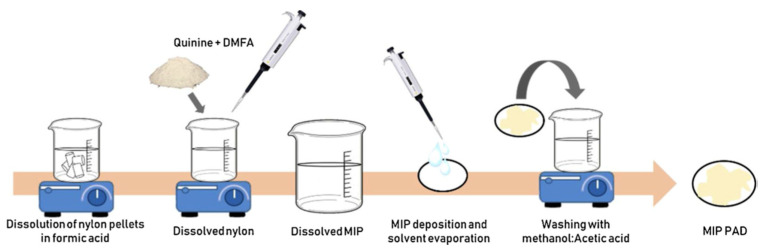
Diagram of the different steps of the synthesis of molecularly imprinted polymer PADs, namely, dissolution of nylon pellets in formic acid, addition of quinine and DMFA, MIP deposition and solvent evaporation, and washing with methanol-acetic acid solution and water prior to drying. Reprinted from [61], with permission from Elsevier, 2019.

**Figure 18 ijms-24-01915-f018:**
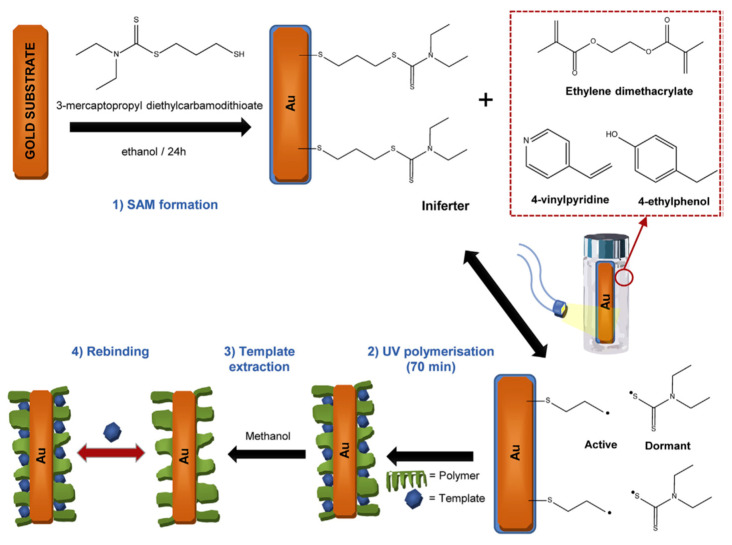
Schematic illustration of the followed whole procedure to develop the MIP sensor. (**1**) Immobilization of the thiol iniferter on a gold substrate. Immersion of the iniferter-coated gold microelectrode in the polymerization solution. (**2**) Coating of the gold microelectrode with an MIP thin layer. Polymerization was initiated by UV radiation. (**3**) Template molecule extraction using methanol. (**4**) Template rebinding was avoided by replacing the extracting medium every 5 min. Reprinted from [64], with permission from Elsevier, 2018.

**Table 1 ijms-24-01915-t001:** Summary of the applications of molecularly imprinted polymers in Spain in recent years.

Group	Type	Template	Matrix	FM	CL	T.P.	D.L.	λexc, λem	Means of Measure	Validation Method	Ref.
**Pesticides**	MISPE	PPZ	Agricultural soil	MAA	DVB-80	PP	LOD: 3.4–3.6 ng g^−1^	-	Toluene	HPLC-DAD	[33]
							LOQ: 1.3–7.6 μg·L^−1^				
	MISPE	Triazinic	Water	MAA y HEMA	EGDMA	PP	LOD: 0.16–0.5 µg L^−1^	λexc = 220 nm	Water	HPLC-DAD	[34]
	SENSOR	Triazine herbicides	Peppermint and tea	MAA	EGDMA	MP	LOD: 0.4–2.5 μg·L^−1^	-	-	HPLC-UV	[35]
	MISPE	Meth-Carb	Water	Acrylamide	EGDMA	NC	LOD: 0.27 µg L^−1^	λexc = 280 nm	Water	FLD	[36]
	SENSOR	Chlorp.	Drinking water samples	Pyrrole	-	EP	LOD: 1 × 10^−7^ μg·L^−1^	-	PB pH 7	SBD	[37]
**Mycotoxins**	MISPE	CDHB	Food	1-ALPP	TRIM	BP	-	λexc = 270 nm λem = 440 nm	MeOH/ACN	HPLC-FLD	[38]
	SENSOR	TeA	Food	DEAM or allyl acetoacetate	EGDMA	PSB	LOD: 500 µg L^−1^	λexc = 330 nm	ACN	HPLC-DAD	[39]
							LOQ: 1700 µg L^−1^				
	SENSOR	TeA	-	[Ru (dab)2(bim)]^2+^ y MAM	EGDMA	RAFT	-	λexc = 500 nm λem = 700 nm	ACN/ water (95:5 *v*/*v*)	Abs and FLD	[40]
**Antibiotics**	MISPE	Sulfam.	Milk sample	MAA	EGDMA	SP	LOD: 3.0 × 10^−7^ mol L^−1^	-	MeOH/Ac. Acet. (9:1)	SVW	[41]
	MISPE	ENRO	Water and urine	MAA	EGDMA	NC	LOD: 0.1–10 µg L^−1^	λexc = 280 nm	Water	HPLC-UV and HPLC-MS/MS	[42]
	MISPE	ENOX	River water	MAA, TFMAA	EGDMA	PSB	LOD: 10^−4^–7 × 10^−4^ µg L^−1^	λexc = 280 nm λem = 440 nm, 515 nm	HEPES 0.05 M pH 7.5	HPLC-FLD	[31]
							LOQ: 4 × 10^−4^–1.5 × 10^−3^ µg L^−1^				
	MISPE	Spir.	Sheep milk	MAA	EGDMA	BP	LOQ: 24.1 µg kg^−1^	λexc = 231 nm	CAN	HPLC-DAD	[43]
**Illicit drugs**	MISPE	Cannab.	Urine	EGDMA	DVB	PPpm	LOD: 0.032–0.748 µg L^−1^	-	BS pH = 5	HPLC-MS/MS	[44]
							LOQ: 0.107–2.50 µg L^−1^				
	MISPE	Cath.	Urine	EGDMA	DVB	PPpm	LOD: 0.32 µg L^−1^	-	PB pH = 5	HPLC-MS/MS	[45]
							LOQ: Athylone = 1.08 µg L^−1^ 3-MMC = 1.18 µg L^−1^				
	MISPE	EME	Water	MAA	EGDMA	ThP	LOD: 3 μg·L−1	-	MeOH: Ac. Acet. (99:1)	UHPLC-MS/MS	[46]
							LOQ: 10 μg·L^−1^				
	SENSOR	Cocaine and meth	Saliva samples	-	-	Nylon-6-P	LOD: 3 μg·L^−1^	-	Water	HPLC-MS/MS	[47]
							LOQ: 10 μg·L^−1^				
**Explosives**	SENSOR	DNP	-	MAA	EGDMA	ThPP	LOD: DNP = 0.59 µmol L^−1^ TNT = 0.29 µmol L^−1^	-	PB pH = 7	VD	[49]
	SENSOR	3-NT and NM	-	-	-	PCL	LOD: 1.59 μg·L^−1^	λexc = 511 nm	Organic solvent	PL	[50]
**Biological** **analytes**	SENSOR	MDA-DAN	Chicken serum	-	-	MGO@MIPy	LOD: 0.003 μM	-	ACN	DVP	[51]
							L-LOQ: 0.01 μM				
	SENSOR	Lysozyme	Chicken egg and commercial drug	-	-	PPy	LOD: 9 × 10^−6^ μg·L^−1^	-	DDW	CV	[52]
							LOQ: 9 × 10^−4^ μg·L^−1^				
**Other analytes**	MISPE	4-NP	Water	4-VP	EGDMA	BP	-	λexc = 260 nm	Water pH = 2	HPLC	[53]
	SENSOR	4-EP or 4-EG	Gallic acid and quercitine	DVB	EGDMA	Co-P	LOD: 4-EP: 1330 μg·L^−1^ 4-EG: 1550 μg·L^−1^	-	PB pH 7.0	Bio-T	[54]
	SENSOR	CDNB	Water	MAA	EGDMA	MagP	LOD: 1200 µg L^−1^	-	BS	DPV	[55]
							LOQ: 4100 µg L^−1^				
	MISPE	DBP	Tea samples	MAA	EGDMA	PP	LOD: 0.92 μg·L^−1^	-	MeOH	HPLC-ESI-MS	[56]
							LOQ: 3.07 μg·L^−1^				
**Contaminant** **removal**	MISPE	DCF and IDM	-	AT	EGDMA	BP	-	λexc = 260 nm	ACN/ Water (5% *v*/*v*)	UVs	[57]
	MISPE	DCF	-	AT	EGDMA	BP	-	λexc = 280 nm	ACN/ Water (5% *v*/*v*) pH = 5–7	UVs	[58]
**Proteins** **purification**	MISPE	DYKD	-	EAMA	EGDMA	PSB	-	λexc = 587 nm λem = 610 nm	TB pH 7.5	CLARIOstar	[59]
	SENSOR	DYKDC	-	EAMA	EGDMA	HP	-	λexc = 220 nm	HEPES	HPLC-DAD	[60]
**New** **polymerization methods**	SENSOR	Quinine	Soda drink	-	-	Nylon-6-P	LOD: 370 µg L^−1^	λexc = 270 nm	Water	FLD	[61]
							LOQ: 1240 µg L^−1^				
	SENSOR/MISPE	R123	-	MAA	TRIM	MagP	-	λexc = 510 nm	ACN	HPLC-UV	[62]
	MISPE	Sulfon.	Water	MAA	EGDMA	Core–shell	LOD: 60 µg L^−1^	λexc = 538 nm	Water	Abs	[63]
							LOQ: 200 µg L^−1^				
	SENSOR	4-EP	-	4VPy and EDMA	EDMA	LP	LOD: 0.30 µM	-	BS pH 10	DPV	[64]
	SENSOR	Gelatin, collagen, and elastin proteins	-	-	-	Elec-P	-	-	BSA	WCA	[65]

Abs: Absorption spectrometry, ACN: Acetonitrile, Ac. Acet.: Acetic acid, AT: Allylthiourea, 1-ALPP: 1-Allylpiperazine, Bio-T: Bioelectronic tongue, BP: Bulk polymerization, BSA: Bovine serum albumin, BS: Buffer solution, Cannab.: Cannabinoids, Cath.: Cathinones, CLARIOstar: CLARIOstar microplate reader from BMG LabTech, CDHB: Cyclododecyl 2,4-dihydroxybenzoate, CDNB: 1-Chloro-2,4-dinitrobenzene, CL: Cross-linker, Chlorp: Chlorpyrifos, Co-P: Co-polymerization, CR%: Cross-reactivity, CV: Cyclic voltammetry, DAD: Diode-Array Detection, DBP: Dibutyl phthalate, DCF: Diclofenac, DDW: Doubly deionized water, DEAM: Diethyl allylmalonate, D.L.: Detection limits, DNP: 2,4-Dinitrophenol, DPV: Differential pulse voltammetry, DR: Dynamic range, DVB: Divinylbenzene, DYKD: Aspartic acid-tyrosine-lysine-aspartic acid peptide, DYKDC: Aspartic acid-tyrosine-lysine-aspartic acid-glycine Peptide, EAMA: N-(2-aminoethyl)methacrylamide hydrochloride, EGDMA: Ethylene dimethacrylate, EDMA: Ethylene glycol dimethacrylate, 4-EG: 4-ethylguaiacol, EME: Ecgonine methyl ester, ENOX: Enoxacin, ENRO: Enrofloxacin, EP: Electropolymerized, Elec-P: Electrospinning, 4-EP: 4-Ethylphenol, FLD: Fluorescence spectrometry, FM: Functional monomer, HEMA: 2-Hydroxyethyl methacrylate, HP: Hierarchically polymerization, HPLC: High-performance liquid chromatography, HPLC-ESI-MS: High-performance liquid chromatography electrospray ionization mass spectrometry, IDM: Indomethacin, LOD: Limit of detection, LP: Living polymerization, MAA: Methacrylic acid, MAM: Methacrylamide, MDA-DAN: Malondialdehyde-Diaminonaphtalene, MeOH: Methanol, Meth: Methamphetamine, Met-carb: Methyl-carbamate, Measur.: Measuring, MISPE: Molecularly imprinted solid-phase extraction, 3-MMC: 3-Methylmethcathinone, MagP: Magnetic polymers, MGO@MIPy: Magnetic graphene molecularly imprinted polypyrrole polymer, MP: Monolithic polymerization, NC: Noncovalent, NM: Nitromethane, 4-NP: 4-Nitrophenol, 3-NT: 3-Nitrotoluene, Nylon-6-P: Nylon-6 pellets polymerization, PB: Phosphate buffer, PCL: CsPbBr3 NCs embedded in a polycaprolactone, PL: Photoluminescence, PP: Precipitation polymerization, PPpm: Polypropylene porous membrane, Py: Pyrrole, PPy: Polymerization of pyrrole, PPZ: Propazine, PSB: Polymerization into silica beads, RAFT: Reversible addition-fragmentation chain-transfer polymerization, R%: Reactivity, R123: Rhodamine 123, SBDs: Smartphone-based detections, SMX: Sulfamethoxazole, SP: Spherical polymerization, Spir.: Spiramycin, Sulfam.: Sulfamethazine, Sulfon.: Sulfonamide, SVW: Square wave voltammetry, TB: Tris buffer, TeA: Tenuazonic acid, TFMAA: 2-Trifluoromethacrylic acid, TNT: 2,4,6 Trinitrotoluene, ThPP: Thermal precipitation polymerization, ThP: Thermal polymerization, T.P.: Type of polymerization, TRIM: Trimethyl trimethacrylate, UV: Ultraviolet, UVs: UV spectrophotometry, 4VPy/4-VP: 4-Vinylpyridine, VD: Voltammetric detection, WCA: Water contact angle.

## Data Availability

The data presented in this study are available upon request from the corresponding author.

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
