# Peer review of "Current Trends in Molecular Imprinting: Strategies, Applications and Determination of Target Molecules in Spain"

_ijms, 2023, doi:10.3390/ijms24031915_

Round 1

Reviewer 1 Report

This paper gathers the information about the main applications and strategies used for the determination of important target molecules using MIPs in Spain in recent years. In my opinion, the manuscript is well organized and it is relevant to the field, and it merits publication after some minor changes

1.Introduction

- Page 3. Lines 68-76.  In this paragraph, some references should be introduced, e.g.  when mentioning that a UV-light source can be used for polymerization at 4ºC in the case of heat-sensitive templates

- The introduction could be shortened. Tables 1-3 can be eliminated, and the names of the main monomers, cross linkers and initiators can be mentioned in the text.

- c) Spherical-nanowire particles

The authors focus on the synthesis of spherical particles and give no details about nanowires or similar materials.

-d) Analytical separations, and Table 4.

The authors mention that all the papers included in the review are summarized in Table 4.  Please, include in the text a brief explanation of the information displayed in the different columns of the table, and about the order used to list the references. This order is not the same than in the manuscript for the different applications (pesticides, mycotoxins, antibiotics, illicit drugs, etc), and therefore, it makes more difficult to look for the information at the same time that you are reading the text.

In the table, “Recognition media” could be more adequate than “Means of measure”

2. Recognition and determination of target analytes

-2.1 Pesticides

Page 8. Lines 219-220. “In this way, two sensing devices… to the former one”.  Please, introduce the reference citations needed to improve the understanding of that phrase.

-2.5 Explosives

Page 15. Lines 439-440. Introduce the limit of detection for TNT in the text

5. New polymerization methods

- Figure 18. The figure caption should be more descriptive of the text

Author Response

This paper gathers the information about the main applications and strategies used for the determination of important target molecules using MIPs in Spain in recent years. In my opinion, the manuscript is well organized and it is relevant to the field, and it merits publication after some minor changes.

Thanks for the comments. The original manuscript has been modified after your suggestions as follows.

1.Introduction

- Page 3. Lines 68-76.  In this paragraph, some references should be introduced, e.g.  when mentioning that a UV-light source can be used for polymerization at 4ºC in the case of heat-sensitive templates

A reference has been added.

- The introduction could be shortened. Tables 1-3 can be eliminated, and the names of the main monomers, cross linkers and initiators can be mentioned in the text.

Tables 1-3 have been eliminated and the names of the main monomers, cross linkers and initiators are now mentioned in the text.

- c) Spherical-nanowire particles

The authors focus on the synthesis of spherical particles and give no details about nanowires or similar materials.

The referee is right. However, the procedures using spherical particles and nanowires usually are identical. Only changes the “core” material where the polymer is grown. That is the reason we have not added extra information.

-d) Analytical separations, and Table 4.

The authors mention that all the papers included in the review are summarized in Table 4.  Please, include in the text a brief explanation of the information displayed in the different columns of the table, and about the order used to list the references. This order is not the same than in the manuscript for the different applications (pesticides, mycotoxins, antibiotics, illicit drugs, etc), and therefore, it makes more difficult to look for the information at the same time that you are reading the text.

Changed

In the table, “Recognition media” could be more adequate than “Means of measure”

Changed

  1. Recognition and determination of target analytes

-2.1 Pesticides

Page 8. Lines 219-220. “In this way, two sensing devices… to the former one”.  Please, introduce the reference citations needed to improve the understanding of that phrase.

Reference citations have been included.

-2.5 Explosives

Page 15. Lines 439-440. Introduce the limit of detection for TNT in the text

The limit of detection for TNT has been added.

  1. New polymerization methods

- Figure 18. The figure caption should be more descriptive of the text

A brief explanation of the process has been included.

Reviewer 2 Report

The manuscript is a review summarizing research advances in molecular imprinting done in Spain. The review is generally well written, however I have some remarks that must be addressed before the final publication:

·         The review focuses on research done in Spain, so it should be mentioned both in title and in the abstract.

·         In abstract the authors clam: “These synthetic polymers are relatively easy to prepare and can be an excellent choice in the substitution of antibodies or enzymes in different kinds of assays.” I disagree here, as antibodies or enzymes cannot be easily substituted. They cannot be used in ELISA test etc. I would modify the sentence.

·         Line 25: should be “processes in which”. There are more mistakes in English in the text, this is just an example. Please proofread it before resubmission.

·         Line 37: text refers only to methacrylic and acrylic MIPs, whereas other MIPs are also possible. I would either remove information regarding radical polymerization or mention that it refers to these MIPs (as they are mostly used in research).

·         Figure 1: should be “molecular imprinting process”.

·         Line 50: I am not sure if I would call it a derivatization. This term reserved for analytical techniques, not synthesis.

·         Line 74: are you sure it should be “whose”?

·         Table 2 and 3: The structures are copied from various sources instead of drawn. It looks unprofessional and ugly.

·         Line 391: is “divynilbencene” correct?

·         Most figures: there is no information regarding the permission for publisher to reuse the material and no information regarding the license.

Author Response

The manuscript is a review summarizing research advances in molecular imprinting done in Spain. The review is generally well written, however I have some remarks that must be addressed before the final publication:

Thanks for the comments. The original manuscript has been modified after your suggestions as follows.

  • The review focuses on research done in Spain, so it should be mentioned both in title and in the abstract.

Title and abstract have been modified.

  • In abstract the authors clam: “These synthetic polymers are relatively easy to prepare and can be an excellent choice in the substitution of antibodies or enzymes in different kinds of assays.” I disagree here, as antibodies or enzymes cannot be easily substituted. They cannot be used in ELISA test etc. I would modify the sentence.

These authors do not think antibodies or enzymes can be easily substituted, only that MIPs are easy to prepare. In ELISAs assays MIP can substitute one of the two antibodies used. Anyway, the sentence has been slightly modified.

  • Line 25: should be “processes in which”. There are more mistakes in English in the text, this is just an example. Please proofread it before resubmission.

“Processes by which” has been modified by “processes in which”. The manuscript has been revised.

  • Line 37: text refers only to methacrylic and acrylic MIPs, whereas other MIPs are also possible. I would either remove information regarding radical polymerization or mention that it refers to these MIPs (as they are mostly used in research).

We agree on that. The text has been changed.

  • Figure 1: should be “molecular imprinting process”.

“Imprinted” has been modified by “imprinting”.

  • Line 50: I am not sure if I would call it a derivatization. This term reserved for analytical techniques, not synthesis.

We disagree a little bit on that. Anyway, the term derivatization has been omitted as the referee’s suggestion.

  • Line 74: are you sure it should be “whose”?

The sentence has been changed.

  • Table 2 and 3: The structures are copied from various sources instead of drawn. It looks unprofessional and ugly.

Following the recommendation of the other reviewer, tables 2 and 3 have been eliminated.

  • Line 391: is “divynilbencene” correct?

No, it is not correct. “Divynilbencene” has been modified by “divinylbencene”.

  • Most figures: there is no information regarding the permission for publisher to reuse the material and no information regarding the license.

Information regarding the permission for publisher to reuse the figures has been added.  

Round 2

Reviewer 2 Report

The Authors have addressed all the comments and corrected their manuscript, thus in my opinion it can be accepted for publishing.